# The Effects of Exogenous Benzoic Acid on the Physicochemical Properties, Enzyme Activities and Microbial Community Structures of *Perilla frutescens* Inter-Root Soil

**DOI:** 10.3390/microorganisms12061190

**Published:** 2024-06-13

**Authors:** Tongtong Xue, Yuxin Fang, Hui Li, Mengsha Li, Chongwei Li

**Affiliations:** 1Engineering Research Center of Agricultural Microbiology Technology, Ministry of Education, Heilongjiang University, Harbin 150500, China; 1577450939@163.com (T.X.); fyx_1107@163.com (Y.F.); lihui920120@163.com (H.L.); 2Heilongjiang Province Key Laboratory of Ecological Restoration and Resource Utilization for Cold Region, School of Life Sciences, Heilongjiang University, Harbin 150080, China; 3Institute of Nature and Ecology, Heilongjiang Academy of Sciences, Harbin 150040, China

**Keywords:** perilla, BA, microbial community structure, physicochemical indicators

## Abstract

This study analyzed the effects of benzoic acid (BA) on the physicochemical properties and microbial community structure of perilla rhizosphere soil. The analysis was based on high-throughput sequencing technology and physiological and biochemical detection. The results showed that with the increase in BA concentration, soil pH significantly decreased, while the contents of total nitrogen (TN), alkaline nitrogen (AN), available phosphorus (AP), and available potassium (AK) significantly increased. The activities of soil conversion enzymes urease and phosphatase significantly increased, but the activities of catalase and peroxidase significantly decreased. This indicates that BA can increase soil enzyme activity and improve nutrient conversion; the addition of BA significantly altered the composition and diversity of soil bacterial and fungal communities. The relative abundance of beneficial bacteria such as *Gemmatimonas*, *Pseudolabrys*, and *Bradyrhizobium* decreased significantly, while the relative abundance of harmful fungi such as *Pseudogymnoascus*, *Pseudoeurotium*, and *Talaromyces* increased significantly. Correlation analysis shows that AP, AN, and TN are the main physicochemical factors affecting the structure of soil microbial communities. This study elucidates the effects of BA on the physicochemical properties and microbial community structure of perilla soil, and preliminarily reveals the mechanism of its allelopathic effect on the growth of perilla.

## 1. Introduction

*Perilla frutescens* Britt., belonging to the family Lamiaceae and genus *Perilla*, was one of the first medicinal and edible homologous plants used in traditional medicine [1]. *Perilla* seeds are rich in various unsaturated fatty acids, such as α-linolenic acid, linoleic acid, and oleic acid, and have effects on regulating blood lipids, improving memory, and anti-inflammatory processes, as well as anti-aging, anti-allergic, and anti-tumor effects [2]. They are known as the “golden plant brain” in the field of nutrition and have an extremely high development value [3]. Heilongjiang Province has the largest planting area and annual export volume of perilla in China, where it is exported to Japan, South Korea, and other places year-round [4]. In recent years, increasingly intensive perilla cultivation in Heilongjiang Province, field management issues, and continuous cultivation have led to obstacles relating to the continuous cultivation of perilla. This issue is increasing in severity, resulting in a continuous decline in the quality and yield of perilla seeds [5], which has become the main bottleneck restricting the development of the perilla industry [6].

By studying the problem of crop succession disorder in depth, it has been found that the occurrence of crop succession disorder can be mainly attributed to the self-toxic substances from plant root secretions and residue decomposition entering the soil and jeopardizing the growth and development of crops [7] and the number of soil-borne disease-causing microorganisms increasing with the number of beneficial microorganisms decreasing [8]. A wide range of chemosensitizers, including BA, p-hydroxyBA, butyric acid, ferulic acid, vanillic acid, cinnamic acid, and rosmarinic acid, have been implicated in autotoxicity [9]. For different species of plants, the types of key autotoxic substances vary [10]. Zhu Wei et al. showed that BA was the main chemosensitizer in peach seedling soil, disrupting the antioxidant enzyme metabolism and synthesis pathway of peach seedlings, altering the morphology of the nucleus and mitochondria, and decreasing the growth index of peach seedlings [11]. Zhou and Wu showed that ferulic acid (a derivative of cinnamic acid) in soil inhibited the growth of cucumber seedlings by altering the structure of soil microbial communities, elucidating the ability of chemosensory substances to influence the diversity of microorganisms [12]. Cui et al. explored the effects of different concentrations of 2,4-di-tert-butylphenol (2,4-DTBP) on the structure of inter-root soil microbial communities of lilies in Lanzhou, and the results showed that the relative abundance of soil fungal pathogens increased and the relative abundance of beneficial microorganisms decreased, increasing the risk of plant diseases developing [13]. To summarize, root secretions not only directly damage plant roots but also affect normal plant development by altering the structure of the host soil microbial community [14]. It can be argued that root secretions are the root cause of the occurrence of crop succession disorder [15]. However, with an increased understanding of some typical allelopathic chemicals and improved means of identifying allelopathic chemicals and microbial community structures in soil, efforts should now be made to understand the impact of allelopathic chemicals on microorganisms in soil environments [16].

BA belongs to the group of phenolic acid chemical substances and is a common aromatic compound in most agricultural soils. Due to its poor biodegradability, it accumulates year after year in the same field or plant soil and is a representative plant allelopathic substance [17]. The impact of BA on the structure and activity of soil microbial communities has two aspects: firstly, it changes the diversity and structural composition of microbial communities; secondly, it changes the biomass and activity of certain microorganisms. For example, Wang et al. found that BA can induce dynamic responses in rice soil microorganisms, alter soil microbial population and community structures, and increase soil fungal diversity [18]; Ding et al. found that BA significantly affects the richness and evenness of rhizosphere microorganisms, leading to changes in microbial community structure and composition [19]. Zwetsloot et al. reported that BA alters the absolute abundance of bacteria and fungi by providing microbial carbon and energy sources [20]. In peanut cultivation, the supply of BA can significantly increase the relative abundance of bacteria and fungi in the soil, reduce the bacteria-to-fungi ratio, and stimulate or inhibit the abundance of several exclusive soil-borne pathogenic microorganisms [21]. It can be inferred that the effect of BA on soil microbial biomass and activity is related to the host plant species, and the promotion or inhibition effects of different concentrations of BA may be completely opposite. For example, phenolic compounds in japonica rice plots reduced soil microbial community biomass but did not decrease soil microbial activity [21]. Chen et al. found that in continuously cultivated, monoculture cucumber soil, BA increased fungal biomass and activity, leading to an increase in pathogenic bacteria and an outbreak of cucumber wilt disease [22]. However, Zheng et al. found that BA has an inhibitory effect on fungal growth [23]. These differences may be caused by differences in phenolic compounds and plant species, as well as differences in their interactions. Due to differences in BA concentration and plant species, there is no unified conclusion on the impact of BA on soil microbial community structures and activity. Thus far, the interaction between perilla plants and soil microbial communities mediated by allelopathic substances is still unclear. The preliminary research group detected a certain concentration of BA in the rhizosphere soil of perilla using high-performance liquid chromatography. Therefore, in this study, BA was added via soil mixing during perilla cultivation to investigate the effect of BA when used as an allelopathic substance on the nutritional composition and microbial community structure of perilla rhizosphere soil. The aim was to elucidate the allelopathic effect of BA on aboveground perilla growth and underground soil nutrient and rhizosphere microbiota. We used physiological and biochemical detection methods and high-throughput sequencing technology to determine soil physicochemical properties, microbial diversity, and function and explored the following issues: (1) the effects of BA concentrations on perilla growth characteristics; (2) the effects of BA concentrations on the soil nutrient, and diversity, composition and function of perilla rhizosphere soil bacterial and fungal communities. This study could give a deep understanding of the allelaphathic effect resulting from BA on perilla growth and the rhizosphere microbiome.

## 2. Materials and Methods

### 2.1. Experimental Material

The experimental soil was collected at the experimental field of Hulan Campus of Heilongjiang University (126°38′43″ E, 46°0′19″ N).

The perilla seeds used for the study were of the Long Perilla 1 variety (a donation from Mr. Wei Guojiang of the Daqing Branch of the Heilongjiang Academy of Sciences).

### 2.2. Experimental Methods

The potting simulation experiment was set up with four different BA treatments: CK (0 mg/kg soil as control), B30 (30 mg/kg), B60 (60 mg/kg), and B90 (90 mg/kg). The setting of BA concentration is based on the pre-experiment conducted by the research group. These treatments had different concentrations, and the treatment was carried out by mixing the soil with 5 kg of soil added to each pot. Each treatment was five replicates.

Perilla seeds with large, full grains were selected, soaked in 2% sodium hypochlorite solution for 20 min, rinsed with distilled water three times, and placed in a box. After 15 d of seedling culture, good seedlings were selected that exhibited similar growth for transplantation into pots accompanied by different concentrations of BA. Three perilla plants were assigned to each pot. During the experiment, the flowerpots were placed outdoors (located in the northern temperate zone with a continental monsoon climate, with an average annual rainfall of 569.1 mm and an average annual temperature of 5.6 °C) and placed randomly. The aboveground part of the plant was harvested 120 days after the growth of perilla and the plant height, root weight, leaf dry weight, stem dry weight, and 100-grain weight were determined.

### 2.3. Soil Sample Collection

The top layer of soil was removed from the perilla pots, and the roots and surrounding soil were also removed by gently shaking the soil from the periphery of the root system, brushing the soil samples adhering to the root surface with a sterile brush, and passing them through a 2 mm stainless steel sieve. Soil samples from the same pot were mixed evenly to obtain a completely homogeneous sample. All of the samples were partly dried naturally for the determination of the physicochemical properties and stored for a short time in a refrigerator at −80 °C for microbial diversity determination.

### 2.4. Determination of Soil Physical and Chemical Indicators

Soil total phosphorus was assayed using a continuous flow analytical system (SKALAR SAN++, Breda, The Netherlands). Soil total potassium was quantified using inductively coupled plasma-atomic emission spectrometry (ICPS-7500, Shimadzu, Japan). Soil total nitrogen was determined using an elemental analyzer (VarioEL III, Langenselbold, Germany) [24]. The potentiometric method was used to examine the soil pH. Organic matter, ammonium nitrogen, and available potassium and phosphorus were determined using a TPY-7PC soil nutrient meter (Hangzhou, China).

### 2.5. Measurement of Soil Enzyme Activity

Soil neutral phosphatase (S-NP) was determined via the colorimetric method using disodium benzene phosphate. Soil urease was determined using the sodium phenol–sodium hypochlorite colorimetric method. Soil catalase (CAT) was determined via potassium permanganate titration. Soil peroxidase (POD) was determined through the use of the 3,3′,5,5′-tetramethylbenzidine (TMB) colorimetric method [25].

### 2.6. Sequencing of Soil Samples

The total DNA of the soil microorganisms was extracted using a Fast DNA SPIN Kit (MP Biomedicals, Irvine, CA, USA), and the concentration was detected via 1% agarose gel electrophoresis and Nanodrop 2000. The Power Clean DNA Clean-up Kit (MoBio, Carlsbad, CA, USA) was used to assess purity. The V3–V4 region of the bacterial 16S rRNA gene was amplified using the universal primers 338F (5′-ACTCCTACGGGGCAGG-3′) and 806R (5′-GACTACHVGGGTWTCTAAT-3′). Targeted amplification of the fungal ITS1 region was performed using the universal fungal primers ITS1F (5′-CTTGGTCATTTAGAGGAAGTAA-3′) and ITS2R (5′-GCTGTGTTCATCGATGC-3′). The amplification conditions were as follows: pre-denaturation at 98 °C for 1 min, denaturation for 10 s for 30 cycles, annealing at 50 °C for 30 s, extension at 72 °C for 30 s, and a final extension at 72 °C for 5 min. After PCR amplification, detection was performed using the QuantiFluor™-ST Blue Fluorescence Quantification System (Shanghai, China). The PCR products were purified using the TAKARA DNA Gel Extraction Kit (TAKARA Biosciences) (Shanghai, China). Subsequently, the Illumina Miseq PE300 sequencer was used for sequencing by the Illumina MiSeq PE300 sequencer. Sequencing was completed by Shanghai (China) Meiji Biopharmaceutical Technology Co., Ltd. (Shanghai, China).

### 2.7. Bioinformatics Process of Sequences

Sequences were processed and analyzed using UPARSE [26]. Non-repetitive sequences were extracted from the optimized sequences after removing single sequences without repeats. Operational taxonomic units (OTU) were clustered sequences without non-repetitive or chimeras according to 97% similarity using USEARCH v11 (http://gfbic5744e4ac1af24badhcunpo9wu0unk6pqv.fiac.hlju.cwkeji.cn/usearch, accessed on 31 March 2024). All sequences were normalized. To obtain taxonomic information on the species corresponding to each OTU, the uclust algorithm (v1.2.22q) was used to taxonomically analyze the OTU representative sequences and count each sample’s community composition at each taxonomic level. The 16s bacterial ribosome database (Silva Release 138.1, http://gfbic12ff13370fea47echcunpo9wu0unk6pqv.fiac.hlju.cwkeji.cn, accessed on 31 March 2024) was used to annotate the taxonomy of each bacterial sample. For fungi, the Unite 8.0 database for fungi was used to carry out taxonomic classification. Operational taxonomic identities were determined using QIIME by executing the BLAST algorithm against sequences in the Unite 8.0 database [27]. Next, the abundance of each operational taxonomic unit (OTU) per sample and the taxonomy of these OTUs were tabulated.

### 2.8. Statistics and Analysis of Data

A one-way ANOVA was performed using SPSS 21.0 software to conduct a minimum significant difference test on the physicochemical parameters of rhizosphere soil and bacterial and fungal alpha diversity at a threshold value of *p* < 0.05. PcoA diagrams were created by the Major Bioinformatic Cloud (https://cloud.majorbio.com/page/tools/, accessed on 30 March 2024). The composition of the soil bacterial and fungal communities was characterized. Using R (version 3.6.3), a Venn diagram, relative abundance diagram, and cluster analysis diagram were created by “ggplot2” [28]. RDA was used to calculate the correlation between soil physicochemical properties and the relative abundance of rhizosphere soil bacterial and fungal genera and also generated by using (https://cloud.majorbio.com/page/tools/, accessed on 30 March 2024). Bacterial functions were performed by the Major Bioinformatic Cloud (https://cloud.majorbio.com/page/tools/, accessed on 30 March 2024) using OTU tables. Fungal function were performed by Major Bioinformatic Cloud (https://cloud.majorbio.com/page/tools/, accessed on 31 March 2024) using OTU tables.

## 3. Results

### 3.1. Perilla Plant Growth Indicators

*Perilla* plant height, root weight, leaf dry weight, stem dry weight, and 100-grain weight results are shown in Table 1. As shown in Table 1, the plant height, root weight, leaf dry weight, stem dry weight, and 100-grain weight of *Perilla* plants showed a decreasing trend with increasing BA addition. Compared with the CK group, the stem dry weight and 100-seed weight were significantly lower (*p* < 0.05) in the B30 group; the plant height, leaf dry weight, stem dry weight, and 100-seed weight in the B60 group, and the plant height, leaf dry weight, and 100-seed weight of seeds in the B90 group were significantly lower (*p* < 0.05); but the difference with the B60 group was not significant (*p* > 0.05). The root weights of the B30 and B60 groups were not significantly different from those of the CK group (*p* > 0.05), but the root weights of the B90 group were significantly lower than those of the B60 and CK groups (*p* < 0.05). The results showed that the addition of BA was detrimental to the growth and development of *Perilla* plants, and this detrimental effect was more pronounced at higher concentrations. In addition, we found that low doses of BA had a more pronounced effect on seed development and high doses of BA had a greater effect on root weight.

### 3.2. Physicochemical Properties of Inter-Root Soil of Perilla frutescens

BA significantly altered the physicochemical properties of the soil (Table 2). The pH gradually decreased from CK to B90. The content of organic matter (SOM) did not change throughout the entire treatment process. The content of total phosphorus (TP) and total potassium (TK) gradually decreased from CK to B90. The content of total nitrogen (TN), available nitrogen (AN), available phosphorus (AP), and available potassium (AK) also showed significant changes in different treatments, with the order of change being B90 > B60 > B30 > CK.

### 3.3. Inter-Root Soil Enzyme Activities of Perilla frutescens

There was a significant change in the enzyme activity of the rhizosphere soil of *Perilla frutescens* (*p* < 0.05) (Table 3), and the activities of urease (UE) and phosphatase (NAP) gradually increased from CK to B90 in the following order: B90 > B60 > B30 > CK. The activities of catalase (CAT) and peroxidase (POD) gradually decreased from CK to B90 in the following order: CK > B30 > B60 > B90. The results showed that different concentrations of BA could increase the activity of soil invertase and reduce the activities of catalase and peroxidase.

### 3.4. Changes in the Structure of Soil Microbial Communities in the Inter-Root Zone of Perilla frutescens

#### 3.4.1. Microbial Cluster Analysis

A visual analysis of the bacterial community structure using Venn diagrams was conducted and a total of 511,584 bacterial sequences were obtained, which were classified into 5145 OTU numbers (similarity > 97%). The results are shown in Figure 1a. The total number of OTUs in the four experimental groups was 1341, of which 429 were CK-specific, 138 were B30-specific, 130 were B60-specific, and 84 were B90-specific. Following the visual analysis of fungal community structure using Venn diagrams, the diagrams show 392,269 fungal sequences, which were divided into 2319 OTU numbers (similarity > 97%), and the results are shown in Figure 1b. The four experimental groups had a total of 244 OTUs, including 476 OTUs specific to CK, 214 OTUs specific to B30, 210 OTUs specific to B60, and 165 OTUs specific to B90. The analysis showed that the microbial community structure of different treatment groups is similar, but not completely the same.

#### 3.4.2. Changes in Microbial Diversity and Homogeneity of Soil Microorganisms with the Addition of Different Concentrations of BA

The Chao1 and ACE indices were used to characterize species richness in the microbial community, and the Shannon and Simpson indices were used to characterize the species evenness in terms of distribution in the microbial community. The inter-root soil bacterial diversity of *Perilla frutescens* with BA was added, as shown in Table 4. As shown in Table 4, the Chao1, Shannon, and ACE indices were significantly increased (*p* < 0.05) in the treatment group compared to the CK group, but the differences between the groups were not significant (*p* > 0.05); the Simpson index was significantly lower (*p* < 0.05), but the difference between groups was not significant (*p* > 0.05). The results showed that BA treatment increased the species and homogeneity of soil bacterial microorganisms in the inter-root of *Perilla frutescens*, increasing the soil microbial diversity, which was significantly higher in the B30 group than in the CK group (*p* < 0.05) and stable in the B60 and B90 groups.

The inter-root soil fungal diversity of BA-added perilla is shown in Table 5. The Chao1 index was significantly increased (*p* < 0.05) in the treated group compared to the CK group, with a tendency to first increase and then decrease with the increase in BA concentration. The Shannon index was not significantly different from the CK group in the B30 and B60 groups (*p* > 0.05) and was significantly lower in the B90 group (*p* < 0.05); the Simpson index was not significantly different between the groups (*p* > 0.05); and the ACE index was significantly higher than that of the control group (*p* < 0.05), but there was no difference between different treatment groups (*p* > 0.05). The results showed that low-concentration BA treatment could increase the fungal microbial species in the rhizosphere soil of *Perilla frutescens*, with no significant change in evenness (*p* > 0.05), and high-concentration BA significantly reduced the number and evenness of soil fungal microbial species, leading to a decrease in microbial diversity (*p* < 0.05).

PCoA sorting and the ANOSIM test were used for different groups based on an operational classification unit (OTU) β comparison of diversity. As shown in Figure 2, differences between soil samples treated with different concentrations of BA were observed. Three samples from the same treatment were combined together and were more similar than samples from different treatments. Among them, the bacterial and fungal communities in the B30, B60, and B90 soil samples were significantly separated from those in the CK soil samples (*p* < 0.05). In addition, the soil bacterial community structures of B30 and B90 were similar, while the soil bacterial community structure of B60 showed distinct differences (Figure 2a). The soil fungal community structures of the three BA treatments were similar (Figure 2b).

#### 3.4.3. Analysis of Microbial Species Composition

The bacterial communities of the inter-root soils of different treatment groups were analyzed at the phylum level, and the results are shown in Figure 3a. As shown in Figure 3a, the dominant phyla in the soil bacterial community were Proteobacteria (31.75%), Actinobacteriota (23.89%), Acidobacteriota (14.24%), Patescibacteria (7.41%), Chloroflexi (6.29%), and Firmicutes (5.46%). The relative abundance of Proteobacteria in the different treatment groups was significantly lower than that in the CK group (*p* < 0.05), but the difference between groups was not significant (*p* > 0.05). The relative abundance of Actinobacteriota, 23.89% in the different treatment groups, was significantly lower than that in the CK group (*p* < 0.05).The relative abundance of Chloroflexi was significantly increased (*p* < 0.05), and Acidobacteriota was significantly decreased (*p* < 0.05) in the different treatment groups.

The fungal communities of inter-root soils of the different treatment groups were analyzed at the phylum level, and the results are shown in Figure 3b. The dominant phyla in the soil fungal community were Ascomycota (45.25%), Mortierelomycota (13.39%), Basidiomycota (27.34%), and Chytridiomycota (2.42%). Compared with CK, the relative abundance of Ascomycota phylum was not significantly different (*p* > 0.05) in the B30 and B60 groups, and it was significantly increased (*p* < 0.05) in the B90 group. The relative abundance of Mortierelomycota increased significantly (*p* < 0.05), but the difference between the groups was not significant (*p* > 0.05); the relative abundance of Basidiomycota decreased significantly (*p* < 0.05), but the difference between the groups was not significant (*p* > 0.05).

Differences in the bacteria from the different treatment groups were analyzed at the genus level, and the results are shown in Figure 4a–c. From Figure 4a–c, it can be seen that there were ten main species of bacterial microorganisms that differed between the treatment groups: *Gemmatimonas*, *Pseudolabrys*, *Bradyrhizobium*, *Pseudonocardia*, *Marmoricola*, *Bryobacter*, *Streptomyces*, *Sphingomonas*, *Nocardioides*, and *Mycobacterium*. Compared with CK, the relative abundance of *Pseudonocardia* was significantly increased (*p* < 0.05), and the relative abundance of *Gemmatimonas*, *Pseudolabrys*, and *Bradyrhizobium* was significantly decreased (*p* < 0.05) in the B30 group, as shown in Figure 4a. The relative abundance of *Marmoricola* increased significantly (*p* < 0.05), and the relative abundance of *Gemmatimonas*, *Pseudolabrys*, and *Bradyrhizobium* all decreased significantly (*p* < 0.05) in the B60 group. The relative abundance of *Marmoricola* was significantly increased (*p* < 0.05) in the B90 group, and the relative abundance of *Marmoricola* and *Nocardia-like* spp. decreased, and the relative abundance of *Bacillus* spp., *Sphingomonas* spp., *Pseudolabilella* spp., and *Chronobacterium* spp. increased with increased BA additions, as shown in Figure 4c.

At the genus level, differential analysis was conducted on the fungi from the different treatment groups, and the results are shown in Figure 4d–f. From Figure 4d–f, it can be seen that there are mainly eight different fungal microorganisms among the different treatment groups: *Mortierella*, *Fusarium*, *Pseudogymnoascus*, *Gibberella*, *Pseudoeurotium*, *Talaromyces*, *Solicocozyma*, and *Penicillium*. Compared with CK, the relative abundance of *Mortierella*, *Fusarium*, *Pseudogymnoascus*, *Pseudoeurotium*, and *Talaromyces* in the B30 group significantly increased (*p* < 0.05), while the relative abundance of *Gibberella*, *Solicocozyma*, and *Penicillium* decreased significantly (*p* < 0.05). In the B60 group, *Mortierella*, *Pseudogymnoascus*, and *Pseudogymnoascus* increased significantly (*p* < 0.05). The relative abundance of *Pseudoniobium* and *Talaromyces* increased significantly (*p* < 0.05). Compared with B90, we found that with the increase in BA concentration, the relative abundance of *Gibberella* and *Penicillium* decreased significantly (*p* < 0.05), while the relative abundance of *Mortierella*, *Pseudogymnoascus*, *Pseudoniobium*, and *Talaromyces* increased significantly (*p* < 0.05), as shown in Figure 4d–f.

#### 3.4.4. Microbial LEfSe Analysis

A linear discriminant analysis (LEfSe) was used to analyze the differences between the bacterial and fungal communities of the four different treatment groups to identify the microbial species with significant differences in relative abundance among the groups. Linear discrimination was based on LDA > 3.5, and the results are shown in Figure 5 and Figure 6. A total of 197 bacterial biomarkers, 84 fungal biomarkers, 37 differential bacteria, and 51 differential fungi were obtained in relation to the different taxonomic units. The presence of *Acidobacteriales*, *Acidobacteriaceae*, *Bradyrhizobium*, *Micropepsaceae*, *Elsterales*, *Frankiales*, *Parcubacteria*, *Occallatibacte*, and *Saccharimonadales* was confirmed along with 25 differential bacterial biomarkers with *Halenospora*, *Filobasidiales*, *Patinella*, *Naganishia*, *Dermateaceae*, *Piskurozymaceae*, *Solicoccozyma,* and *Paraglomeraceae* and 32 differential fungal biomarkers with substances. In total, 4 differential bacterial biomarkers, including *Anaerolineaceae*, *Nitrosomonadaceae*, and *BIrii41,* and 7 differential fungal biomarkers, including *Sordariomycetes*, *Glomerellales*, and *Lasiosphaeriaceae*, were present in B30. The *Marmoricola* differential bacterial biomarker was identified in B60 with 6 differential fungal biomarkers, such as *Sordariales* and *Hypocreales*. In total, 7 differential bacterial biomarkers, such as *Microtrichales*, *Gemmatimonadaceae*, *Iamiaceae*, *SC-I-84*, and *Comamonadaceae*, were present in B90, along with 7 differential fungal biomarkers, such as *Ramophialophora* and sedis.

#### 3.4.5. Correlation between Microorganisms and Soil Physicochemical Properties

The correlation analysis (RDA) between the dominant bacterial and fungal genera in rhizosphere soil and soil physicochemical indicators is shown in Figure 7. The cumulative changes in the first and second RDA axes of bacterial microorganisms were 36.13% and 10.31%, respectively (Figure 7a), while the cumulative changes in the first and second RDA axes of fungal microorganisms were 25.13% and 10.18%, respectively(Figure 7b), which demonstrated that the structure of the bacterial community and the fungal communities was substantially influenced by the physicochemical properties of the soil.

AP, TN, and AN were positively correlated with the relative abundance of the bacterial genera *Marmoricola*, *SC-I-84*, Acidobacteriales, *Gemmatimonadaceae,* and *Pseudarthrobacter* and negatively correlated with *Pseudolabrys*, *Gaiellales*, and *Gemmatimonas*. TK was positively correlated with the relative abundance of *Gaiellales* and *Pseudolabrys* and negatively correlated with *Marmoricola*. *Marmoricola* and *SC-I-84* abundances were negatively correlated with pH, whereas those of *Pseudolabrys* and *Gaiellales* were positively correlated (Table 6, *p* < 0.05). AK was the most important physicochemical indicator affecting the relative abundance of bacteria.

Ak, TN, and AN were positively correlated with the relative abundance of the fungus genera *Pseudombrophila*, *Peziza*, *Tausonia,* and *Pseudeurotium* and negatively correlated with *Sordariomycetes* and *Basidiomycota*. TK was positively correlated with the relative abundance of the fungus genera *Basidiomycota* and negatively correlated with *Cephalotrichum* and *Pseudombrophila*. *Pseudeurotium* and *Cystofilobasidiales* abundances were negatively correlated with pH, whereas those of *Basidiomycota* were positively correlated (Table 7, *p* < 0.05). TK was the most important physicochemical indicator affecting the relative abundance of fungus.

### 3.5. Functional Forecast Analysis

PICRUSt annotates the COG and KEGG functions of each OTU based on its corresponding green ID and, through functional prediction analysis, preliminarily explains the functions of the microbial community in the sample. As shown in Figure 8, the dominant bacterial groups are mainly enriched in relation to amino acid metabolism, carbohydrate metabolism, signal transduction, cell wall biofilm synthesis, and energy metabolism (Figure 8a). Fungal functional classification mainly includes saprophytic organisms, plant pathogens, and animal pathogens. Therefore, it can be inferred that the microbial groups in the BA treatment group are mostly saprophytic organisms, plant pathogens, and animal pathogens.

## 4. Discussion

### 4.1. Effect of BA Addition on the Physicochemical Properties of Perilla frutescens Inter-Root Soil

BA is an important secondary metabolite in the synthesis and decomposition of plant substances, which can enter the soil through root secretion. It is a common plant allelopathic substance [29]. In this study, the addition of BA causes soil acidification (neutral soil with a pH between 6.5 and 7.5; acidic soil, below 6.5) [30], but the differences between different groups are not significant. This may be because benzoic acid itself is acidic (pH 5.8) and usually adsorbs on soil particles [31]. Its adsorption capacity decreases with the decrease in pH, and it is also influenced by the surface distribution and adsorption of organic matter. Not only does it affect soil fertility, but it also affects the absorption of soil nutrients by crops, thereby affecting the growth and development of plants and product quality. The organic matter content in soil is usually stable and can only undergo significant changes over a large time scale. In this experiment, there was no significant difference in soil organic matter content among the different treatment groups (*p* > 0.05). This is consistent with the research findings of Villarino et al. [32]. Nitrogen, phosphorus, and potassium are very important nutrients in soil, playing a crucial role in the growth and development of plants. The mechanism of plant nutrient absorption is that plant roots actively absorb nutrients from the soil through their own physiological mechanisms [33], which require ATP and carrier proteins to assist in transportation. In this study, compared with the CK group, with increased BA addition, the content of total nitrogen (TN), available nitrogen (AN), available phosphorus (AP), and available potassium (AK) increased, indicating that the plant root system is not lacking in nutrients; rather, its ability to absorb and utilize nutrients is reduced. This is consistent with Blum’s report that BA inhibits nutrient absorption in plant roots, leading to growth inhibition [34]. Based on the results of the perilla root weight experiment, we speculate that due to the addition of BA, the pH of the root soil decreases, and the transportation of substances inside and outside the cell membrane slows down, resulting in weakened plant root absorption functions in relation to nutrients, and even peroxidation reactions. The plant roots become smaller, or tissue cells are damaged, and the absorption of nitrogen, phosphorus, and potassium decreases, which is proportional to the concentration of BA. Usually, plant roots can secrete a large amount of disaccharides such as sucrose, promoting the movement of microorganisms in the soil and selectively shaping the unique microbial community of the root system [35]. The inhibition of perilla roots by BA leads to an uneven distribution of sucrose in various parts of the plant, resulting in the poor growth and development of perilla plants and decreased seed production.

### 4.2. Effect of BA Addition on the Inter-Root Soil Enzyme Activities of Perilla frutescens

Soil enzymes mainly come from plant roots and soil organisms and are the main catalysts for the decomposition, turnover, and mineralization of soil organic matter [36]. They directly participate in various physiological and biochemical reactions in the soil [37]. Studies have shown that the forms and contents of soil organic matter, nitrogen, phosphorus, and potassium [38] are all related to soil enzyme activity [39] and are of great significance for plant growth regulation [40]. Urease is mainly present in soil bacteria, fungi, and spores, and it is used to catalyze the hydrolysis of urea into ammonia and carbon dioxide [41]. Phosphatase is a key enzyme in the soil phosphorus cycle that is capable of catalyzing the conversion of insoluble inorganic phosphorus into effective phosphorus in the form of PO_4_^3−^, which is absorbed and utilized by plants. In this study, compared with the CK group, the addition of BA significantly increased the activities of urease and phosphatase (*p* < 0.05). However, with increasing BA addition, there was no significant difference between the groups (*p* > 0.05), indicating that BA promotes increased soil urease activity and converts soil organic nitrogen into available nitrogen that plants can use [42], which is consistent with the increase in soil AN content in this experiment. The addition of BA significantly increased phosphatase activity in the rhizosphere soil of *Perilla frutescens* (*p* < 0.05), and it was positively correlated with the amount of BA added. This indicates that the addition of BA can improve soil phosphatase activity, promote the conversion of soil organic phosphorus into inorganic phosphorus [43], and increase the content of available phosphorus in the soil, which is consistent with the increase in soil AP content in this experiment.

Catalase (CAT) is a type of oxidoreductase produced by microorganisms or plants, which is an important enzyme in soil. It can use hydrogen peroxide as a substrate to quickly convert waste generated by soil metabolism into harmless or less toxic substances, while releasing oxygen, reducing the toxicity of hydrogen peroxide accumulation to soil microorganisms and plant roots [44] and enhancing plant antioxidant capacity; moreover, its activity can reflect the soil respiration intensity [45]. Peroxidases have a strong removal effect on some pollutants or their derivatives in soil, eliminating the toxic effects of pollutants by converting them into other products. Peroxidases can utilize hydrogen peroxide and oxygen in other organic peroxides formed in soil due to microbial activity and the action of certain oxidase enzymes to oxidize soil organic matter, catalyze the oxidation reaction of various aromatic compounds (phenol, substituted phenols, aniline, polycyclic aromatic hydrocarbons, etc.), catalyze the oxidation–reduction reaction between hydrogen peroxide (H_2_O_2_) and iron minerals, oxidize colorless substrates of 3,3′, 5,5′-tetramethylbenzidine (TMB) to blue products (TMBox), and release effluents [46]. Hydrogen peroxide is a very active oxidant, and the decrease in the activity of catalase and peroxidase leads to the accumulation of excessive hydrogen peroxide in the soil, causing damage to soil microorganisms and plants [47]. In this study, the activities of catalase and peroxidase significantly decreased in the group treated with BA (*p* < 0.05), but with increasing BA concentrations, there was no significant difference between the groups (*p* > 0.05), indicating that the addition of BA led to a decrease in the activity of catalase and peroxidase in the root soil, damage to the antioxidant system of perilla, damage to its own immune system, and the poor development or toxicity of perilla roots, leading to decay. The difference between the two enzyme activity groups was not significant, indicating that there is no significant linear relationship between the amount of BA added and the activity of antioxidant enzymes. Further research is needed to investigate the reasons behind this.

### 4.3. Effects of BA Addition on the Structure and Function of Inter-Root Soil Microbial Communities of Perilla frutescens

Soil microorganisms play a crucial role in plant nutrition and health [48], and microbial community diversity is closely related to crop growth and development [49]. The results of this study found that, when compared with the CK group, the Chao1 and ACE indices of bacteria and fungi in the group with added BA were positively correlated with BA concentration. This may be due to the additional carbon source provided by exogenous BA, which stimulates the growth of in situ soil microbial communities and increases the richness of bacteria and fungi. This result is consistent with previous research. For example, Qu et al. studied the effects of two phenolic acids on soil microbial populations and found that phenolic acids can selectively enhance specific microbial populations in the soil [50]. On the other hand, it may also be related to pH changes, which increase the efficiency of organic carbon conversion to biomass carbon and alter nutrient utilization efficiency, physiological metabolic activity, and competition among microbial populations, directly or indirectly affecting the microbial diversity of soil [51]. Rousk et al. proposed that low pH values can reduce microbial indicators of soil quality, such as fungi, bacteria, and microbial biomass and, to a lesser extent, reduce microbial activity without affecting metabolism [52]. Fungi in soil have a strong tolerance for acidification, and Moran et al.’s research has shown that fungal biomass is not affected by pH values [53]. Therefore, the changes in fungal biomass are not as significant as those in bacteria, and the abundance of some acidophilic microbial populations changes significantly, disrupting the distribution balance between bacterial and fungal communities. In this study, the addition of BA reduced soil pH, resulting in weak acidity in the soil, indicating that higher concentrations of hydrogen ions inhibited microbial proliferation, which is consistent with the findings of Sui et al. [54]. Therefore, BA can disrupt the ecological balance of soil microorganisms in the rhizosphere of *Perilla frutescens*, resulting in changes in microbial evenness. As the concentration of BA increases, the complexity and stability of soil microbial communities gradually decrease.

At the phylum level, the composition of bacterial communities is relatively similar. Actinobacteria, Proteobacteria, Acidobacteriota, and Chloroflexi are the phyla with a higher relative abundance in the rhizosphere soil of *Perilla frutescens* under all treatments. This indicates that the addition of BA did not alter the composition of the relative abundance of the main bacterial phyla in the soil. Dagher et al. found that Proteobacteria are widely distributed in soil, and their abundance is easily affected by environmental factors, playing an important regulatory role in nitrogen cycling and related to the accumulation of plant biomass [55]. Nguyen et al. proposed Actinobacteria are acidic microorganisms that decompose organic matter to provide plant cells with the necessary nutrients for growth [56], and these microorganisms have a positive impact on plant disease resistance [57]; Kalam et al.’s research found that Acidobacteria have genes involved in different metabolic pathways, regulating the geochemical cycle, breaking down macromolecular biopolymers, secreting extracellular polysaccharides, and promoting plant growth [58]. Actinobacteria and Acidobacteria are acidophilic microorganisms; therefore, their relative abundance is significantly higher under acidic conditions [59]. In this study, with increasing BA concentration, soil pH significantly decreased from 6.8 to 5.9, and it was found that actinomycetes increased with decreasing pH, which is the main reason for the increase in actinomycetes. However, in this study, it was found that the relative abundance of Acidobacteria was significantly reduced, which may be due to BA stress leading to poor root development in the case of *Perilla frutescens* and reduced colonization sites of Acidobacteria. Osiel’s study on the interaction between Acidobacteria and plants found that Acidobacteria have the ability to form flagella, which drive chemotaxis in plant roots [60].

The results of all of the fungal phyla samples indicate that Ascomycota and *Mortierella* are dominant fungal groups in soil, but the effects of different concentrations of BA on the relative abundance of *Mortierella* vary. The results of this study indicate that B30 and B60 did not significantly change the relative abundance of *Mortierella*, while B90 significantly increased the relative abundance of *Mortierella*. The phylum *Aspergillus* is one of the most abundant and diverse saprophytic fungi in acidic and alkaline soils, establishing extensive beneficial or pathogenic interactions with hosts [61]. Liu et al. found that BA can cause *Fusarium* wilt disease in cucumber during continuous monoculture [62]. In addition, Haichar’s study found that BA can promote the biomass and activity of *Ralstonia solanacearum* in rhizosphere soil, leading to an outbreak of tobacco bacterial wilt disease [63]. BA is clearly the key factor to address when adding pathogenic bacteria to cause the plant diseases considered in this article. Although an increasing trend was observed in the case of Ascomycota and other fungal phyla, there was no significant change. Therefore, it can be inferred that the concentration of BA does not have a significant impact on the relative abundance of fungi in the rhizosphere soil of continuously cropped perilla.

At the genus level, there are 10 different bacterial and fungal species in the groups treated with different concentrations of BA: *Gemmatimonas*, *Pseudolabrys*, *Bradyrhizobium*, *Pseudonocardia*, *Marmoricola*, *Bryobacter*, *Streptomyces*, *Sphingomonas*, *Nocardioides*, and *Mycobacterium*. There are 4 different fungal microorganisms: *Ascomycota*, *Glomeromycota*, *Basidiomycota*, and *Chytridiomycota*. Compared with CK, the addition of BA significantly reduced the relative abundance of these 10 bacterial genera (*p* < 0.05). Dias et al. found in their research that *Gemmatimonas* is a phosphate-accumulating bacterium that can regulate the accumulation of phosphorus and energy conversion during cellular metabolism, promoting plant development and maturation [64]. Zhao et al. found in their research that *Bradyrhizobium* has various biochemical functions, including biological nitrogen fixation and carbon fixation. It belongs to the family Rhizobiaceae and interacts with the roots of leguminous plants to fix nitrogen in the atmosphere, improving the nitrogen availability of plants. It also has an extremely high agricultural value [65]. *Streptomyces* is an important antagonistic microorganism with the ability to produce various antibiotics. Studies have shown that *Streptomyces* can promote plant growth and induce overall plant resistance by secreting multiple secondary metabolites. Maila et al. found that *Streptomyces* produces iron carriers, VOC, and ACC deaminases in tomatoes; furthermore, dissolving phosphate can promote tomato plant growth and improve its disease resistance [66]. Song et al. found in their research that *Sphingomonas* genus can efficiently decompose aromatic compounds and has the ability to reduce the concentration of allelopathic substances [67]. Due to the addition of BA, soil pH and nutrient content decrease, inhibiting the growth and development of perilla, which may be the main reason for the decrease in the abundance of the abovementioned bacterial genera.

However, from the perspective of fungal genera, as the concentration of BA increases, the relative abundance of *Mortierella*, *Pseudogymnoascus*, *Pseudoeurotium*, and *Talaromyces* all significantly increase (*p* < 0.05). Jiang et al. found in their research that *Mortierella* is an oily fungus that can accumulate a large amount of highly unsaturated lipids, often causing plant roots to be unable to effectively absorb nutrients and water, leading to plant diseases such as root rot [68]. Hassan et al. found in their research that *Pseudogymnoascus*, as a saprophytic organism, often exhibits a pathogenic lifestyle [69]; Avontuur et al. found in their research that *Pseudomonas* often causes diseases such as root rot, wilt, and leaf mold [70]. Zhang et al. found that *Talaromyces* is an intracellular pathogenic fungus [71]. The relative abundance of harmful fungal genera significantly increased (*p* < 0.05) in this experiment, and the soil environment deteriorated.

The further annotation of fungal functions reveals that the microbial groups in the BA treatment group mostly involve saprophytic organisms, plant pathogens, and animal pathogens. It can be seen that BA significantly increases the abundance of harmful fungi in the rhizosphere soil of *Perilla frutescens*. The reproduction and accumulation of these pathogenic microorganisms in the soil cause significant decomposition of the organic matter in the soil, leading to plant malnutrition, an increased plant disease index, and decreased crop yield. However, we found that this change is not linearly related to the amount of BA added, indicating that the effect of BA on pathogenic microorganisms in the rhizosphere soil of *Perilla frutescens* is limited.

### 4.4. Correlation between Microorganisms and Physicochemical Indicators in Inter-Root Soils

The results of the redundancy analysis indicate that available phosphorus and available nitrogen are the main factors affecting the composition of bacterial communities, while total nitrogen is the main factor affecting the composition of fungal communities. The content of AP, AN, and TN is significantly positively correlated with the relative abundance of the *Bradyrhizobium*, *Streptomyces*, *Devosia*, *Rhodanobacter*, *Pseudogymnoascus*, *Bryobacter*, *Halenospora*, *Mortierella*, *Peziza*, *Fusarium*, *Talaromyces*, and *Tausonia* fungal genera (*p* < 0.05), and it is significantly positively correlated with the relative abundance of *Halenospora*, *Mortierlla*, *Peziza*, and *Fusaria*. There is a significant negative correlation (*p* < 0.05) between the relative abundance of the bacterial genera *Talaromyces* and *Tausonia* and the fungal genera *Sedogymnoascus*, *Cephalotrichum*, *Schizothecium*, and *Pseudomycohila*. *Bradyrhizobium* and *Pseudolabrys* are well-known nitrogen-fixing agents that have a symbiotic relationship with plant roots [72]. They first convert nitrogen into organic nitrogen, and subsequently increase nitrogen levels in rhizosphere soil, thus enhancing effective absorption and the utilization of nitrogen by the host [73]. *Bryobacter* can help to degrade organic matter in soil and maintain the water balance by slowly releasing stored water [74]. *Streptomyces* can produce enzymes, iron carriers, organic acids, and pigments and degrade organic matter and dissolve phosphates. Phosphorus is an important component of functional components such as nucleic acids, nucleotides, and phospholipids. It is an important organic component in plant photosynthesis and respiration, and it is a necessary element for various coenzymes, high-energy phosphate bonds, and oxidative phosphorylation reactions [75]. It has a regulatory effect on the metabolism of sugars, proteins, and fats in plants. At the same time, rhizosphere microorganisms secrete various metabolites (mainly including sugars, organic acids, and amino acids) to the rhizosphere, recruit functional bacteria such as phosphorus-solubilizing bacteria and potassium-solubilizing bacteria to colonize the plant rhizosphere, and secrete soil enzymes such as phosphatase, urease, and catalase, thereby increasing soil enzyme activity and nutrients in the rhizosphere. In this study, based on the annotation of microbial functions, it was found that the role of functional microorganisms is more concentrated in amino acid metabolism, carbohydrate metabolism, and energy metabolism. RDA analysis further revealed that AP, AN, and TN are the main physicochemical factors affecting the composition and distribution of dominant soil microbial communities. Considering the degree of influence, nitrogen is the most critical factor affecting the distribution of microbial communities, and nitrogen is positively correlated with nitrogen-fixing bacteria in the dominant microbial community, which is consistent with the research results of Morales et al. [76].

## 5. Conclusions

Allelopathic substances are secreted by plants and microorganisms which, in turn, have a significant impact on their growth and development. This study measured the effects of different concentrations of BA on the physicochemical properties, microbial diversity, and functions of perilla rhizosphere soil. We found that (1) BA not only causes soil acidification and reduces the active absorption of effective nutrients by perilla but also accelerates the transformation rate of soil nutrients, showing different effects in relation to perilla growth; (2) BA can reduce the relative abundance of beneficial bacteria in rhizosphere soil, such as *Gemmatimonas*, *Pseudomonas*, and *Bradyrhizobium,* and increase the relative abundance of harmful fungi, such as *Pseudomonas*, *Pseudoeurotium*, and *Talaromyces*, causing soil environmental degradation and worsening plant diseases. However, the self-toxic effect of BA is limited. (3) AP, AN, and TN are significantly positively or negatively correlated with the relative abundance of dominant microorganisms in the rhizosphere of *Perilla frutescens*, and they are the main physicochemical factors affecting the composition of microbial community structures in the soil. Benzoic acid has adverse effects on the growth of perilla and has allelopathic effects on its growth. This study elucidated the effects of BA on the growth of perilla from the perspectives of soil physicochemical indicators, enzyme activity, and microbial community structure and function, laying an important theoretical foundation for the further exploration of the mechanisms of action of allelopathic substances. However, complex soil environments introduce many difficulties to the study of exudates, and researchers still need to further study the effects of different root exudates on the rhizosphere microecology of perilla to establish a foundation to further reveal the mechanisms of continuous cropping obstacles in relation to perilla.

## Figures and Tables

**Figure 1 microorganisms-12-01190-f001:**
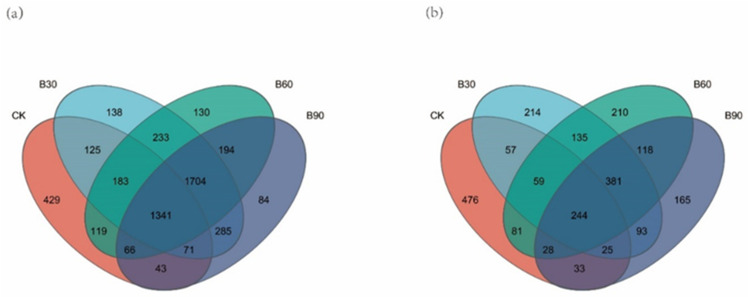
The figure shows a Venn diagram of the bacteria (**a**) and a Venn diagram of the fungi (**b**). Each color in the Venn diagram represents a sample. Overlapping circles represent the number of OTUs shared by the sample; non-overlapping sections represent the number of OTUs specific to the sample.

**Figure 2 microorganisms-12-01190-f002:**
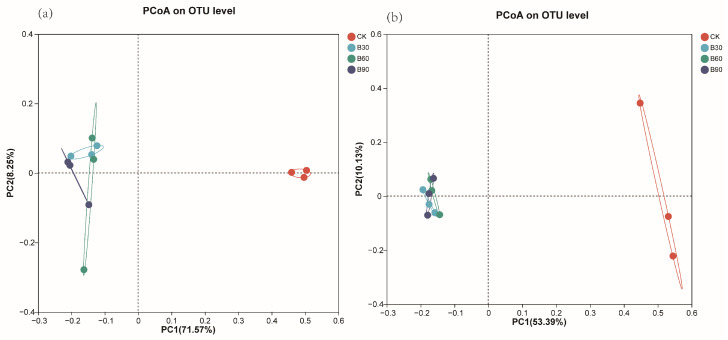
Bacteria (**a**) and fungi (**b**) PCOA in different BA soils.

**Figure 3 microorganisms-12-01190-f003:**
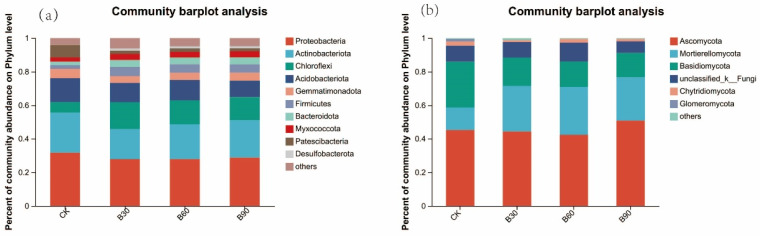
The figure shows the relative abundance of bacteria (**a**) and fungi (**b**) at the phylum level.

**Figure 4 microorganisms-12-01190-f004:**
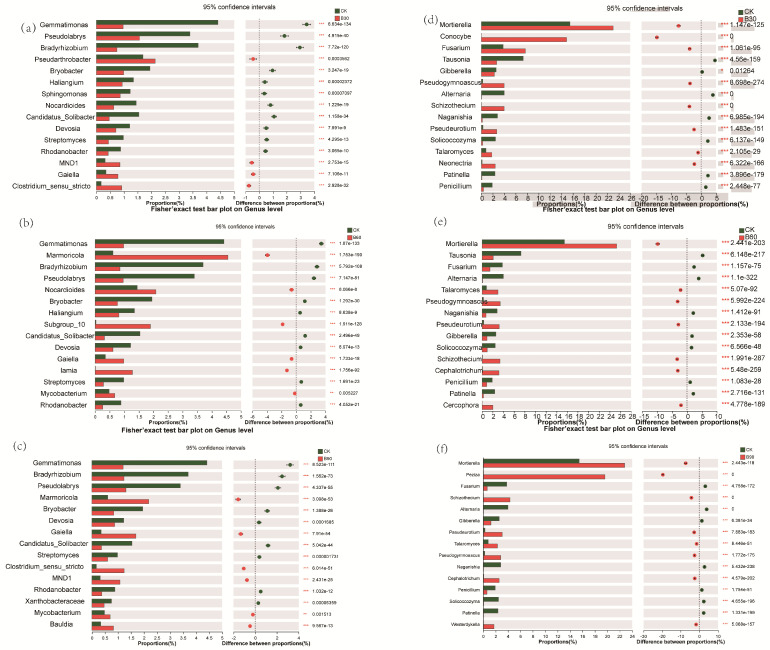
The figure shows the difference in the relative abundance of bacteria CK-B30 (**a**), CK-B60 (**b**), and CK-B90 (**c**) and fungi CK-B30 (**d**), CK-B60 (**e**), and CK-B90 (**f**) in the rhizosphere soil of *Perilla frutescens* at the genus level. On the right is the *p*-value. “*” indicates a significant difference between the two data sets (n = 3, *p* < 0.05); “**” indicates a highly significant difference between the two data sets (n = 3, *p* < 0.01); and “***” indicates a highly significant difference between the two data sets (n = 3, *p* < 0.001), using Welch’s *t*-test.

**Figure 5 microorganisms-12-01190-f005:**
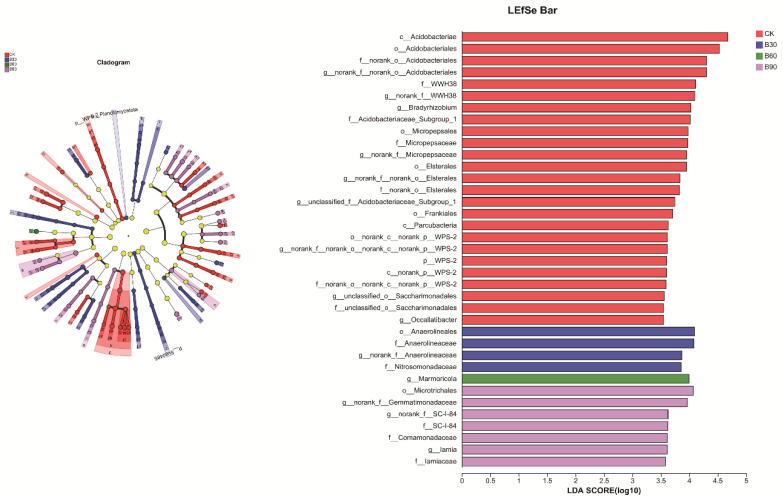
The figure shows a display diagram of differential taxonomic units among bacterial groups in the soil of different treatment groups.

**Figure 6 microorganisms-12-01190-f006:**
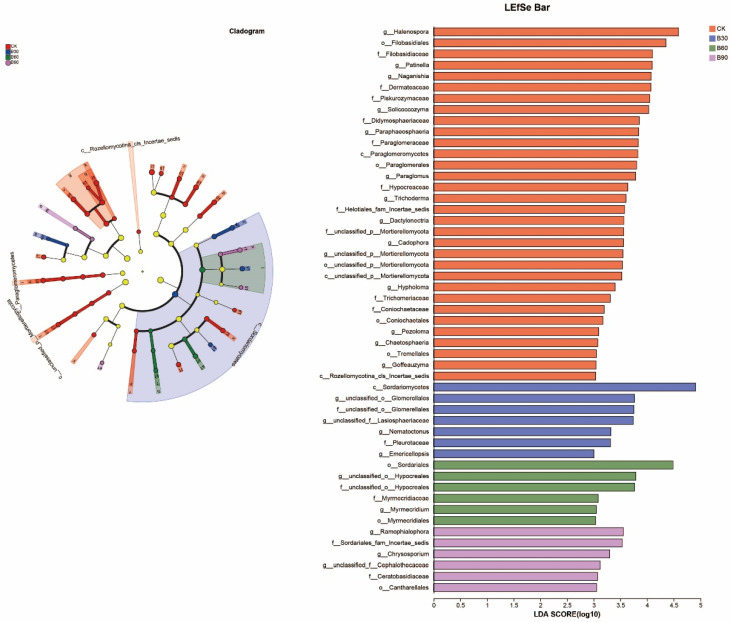
The figure shows a display diagram of differential taxonomic units among fungal groups in the soil of different treatment groups.

**Figure 7 microorganisms-12-01190-f007:**
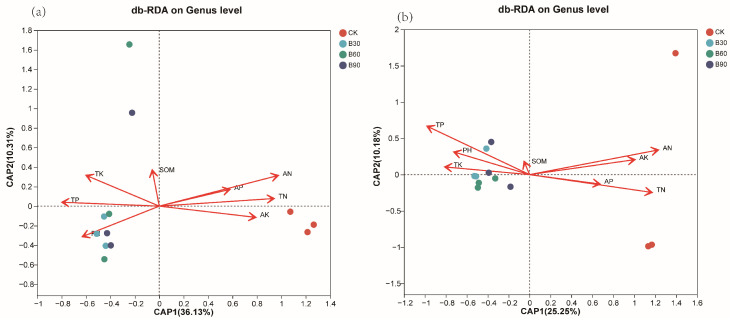
This graph shows the correlation between soil physicochemical properties and bacteria (**a**) and fungi (**b**) at the OTU level.

**Figure 8 microorganisms-12-01190-f008:**
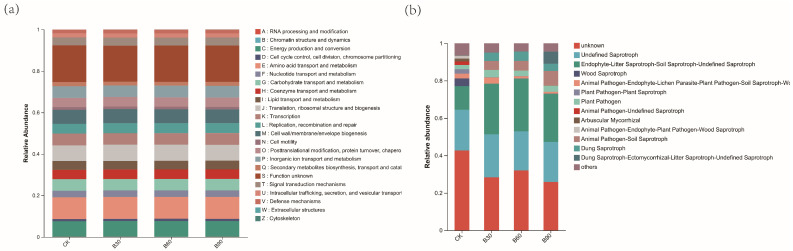
The figure shows (**a**) a box chart of bacterial COG functional classification statistics, note: the abscissa represents the COG secondary functional number, the ordinate represents functional abundance, and (**b**) the FUNGuild functional classification statistics bar chart of fungi, note: the *X*-axis represents different groups, and the *Y*-axis represents the abundance ratio of Guild in different groups.

**Table 1 microorganisms-12-01190-t001:** Growth indicators of perilla plants.

Sample	Plant Height (cm)	Root Weight (g)	Leaf Dry Weight (g)	Stem Dry Weight (g)	Hundred Grain Weight (g)
CK	43.00 ± 2.08 a	1.96 ± 0.57 a	0.13 ± 0.01 a	0.57 ± 0.03 a	0.73 ± 0.01 a
B30	41.00 ± 0.58 a	1.86 ± 0.62 a	0.11 ± 0.01 ab	0.49 ± 0.01 b	0.67 ± 0.01 b
B60	36.67 ± 0.88 b	1.76 ± 0.54 a	0.09 ± 0.01 bc	0.46 ± 0.02 b	0.52 ± 0.01 c
B90	35.00 ± 0.57 b	0.89 ± 0.13 b	0.07 ± 0.01 c	0.39 ± 0.01 c	0.47 ± 0.03 c

Note: The data in the table are the mean ± standard error. Different letters in the same column mean a significant difference among the mean values (*p* < 0.05), the same as below.

**Table 2 microorganisms-12-01190-t002:** The table shows the changes in the physical and chemical properties of the perilla rhizosphere soil.

Sample	pH	SOM (g/kg)	TN (g/kg)	TP (g/kg)
CK	6.80 ± 0.34 a	54.67 ± 11.33 a	1.45 ± 0.05 d	0.74 ± 0.04 ab
B30	6.20 ± 0.06 b	55.00 ± 11.00 a	1.56 ± 0.02 c	0.70 ± 0.01 bc
B60	6.06 ± 0.13 b	55.66 ± 10.33 a	1.64 ± 0.02 bc	0.65 ± 0.02 bc
B90	5.93 ± 0.15 b	55.33 ± 10.66 a	1.74 ± 0.02 ab	0.61 ± 0.04 c
	TK (g/kg)	AN (mg/kg)	AP (mg/kg)	AK (mg/kg)
CK	3.65 ± 0.27 a	43.33 ± 3.33 c	50.33 ± 7.02 c	54.00 ± 7.15 b
B30	3.41 ± 0.27 ab	49.00 ± 3.79 bc	63.00 ± 6.00 b	64.67 ± 2.67 ab
B60	3.05 ± 0.23 b	55.33 ± 3.33 ab	78.00 ± 1.66 a	73.33 ± 3.33 ab
B90	2.91 ± 0.07 b	63.33 ± 2.33 a	78.33 ± 1.66 a	82.00 ± 5.00 a

Note: The data in the table are the mean ± standard error. Different letters in the same column mean a significant difference among the mean values (*p* < 0.05), the same as below.

**Table 3 microorganisms-12-01190-t003:** The table shows the changes in the enzyme activity in the rhizosphere soil of *Perilla frutescens*.

Sample	Soil Enzyme Activity			
	NAP (U·g^−1^)	Urease (U·g^−1^)	CAT (U·g^−1^)	POD (U·g^−1^)
CK	16,460.31 ± 4072 c	697.33 ± 31.64 b	11.22 ± 0.15 a	162.24 ± 10.86 a
B30	21,373.01 ± 1432 b	917.33 ± 38.66 a	8.70 ± 0.23 b	137.56 ± 13.70 b
B60	27,236.11 ± 4000 a	973.61 ± 29.63 a	7.37 ± 0.09 b	125.85 ± 12.57 b
B90	29,751.67 ± 2083 a	1080.53 ± 42.54 a	6.50 ± 0.15 b	110.60 ± 10.03 b

Note: The data in the table represent the mean ± standard error. Different letters in the same column mean a significant difference among the mean values (*p* < 0.05), the same as below.

**Table 4 microorganisms-12-01190-t004:** The table shows the diversity index of bacterial microorganisms.

Sample	Chao1 Index	Shannon Index	Simpson Index	ACE Index
CK	2120 ± 129 b	6.01 ± 0.087 c	0.0065 ± 0.0006 a	2144 ± 124 a
B30	3821 ± 53 a	6.85 ± 0.008 a	0.0028 ± 0.0001 b	3926 ± 108 b
B60	3687 ± 150 a	6.72 ± 0.056 ab	0.0035 ± 0.0005 b	3748 ± 132 b
B90	3643 ± 68 a	6.65 ± 0.038 b	0.0044 ± 0.0008 b	3727 ± 111 b

Note: The data in the table represent the mean ± standard error. Different letters in the same column mean a significant difference among the mean values (*p* < 0.05), the same as below.

**Table 5 microorganisms-12-01190-t005:** Fungal microbial diversity index.

Sample	Chao1 Index	Shannon Index	Simpson Index	ACE Index
CK	596 ± 48 c	4.258 ± 0.083 a	0.048 ± 0.0053 a	597 ± 49 b
B30	754 ± 24 a	4.222 ± 0.113 a	0.043 ± 0.0059 a	756 ± 49 a
B60	780 ± 79 a	4.370 ± 0.166 a	0.033 ± 0.0032 a	782 ± 77 a
B90	703 ± 31 b	3.998 ± 0.071 b	0.050 ± 0.0063 a	707 ± 28 a

Note: The data in the table represent the mean ± standard error. Different letters in the same column mean a significant difference among the mean values (*p* < 0.05), the same as below.

**Table 6 microorganisms-12-01190-t006:** Pearson’s rank correlations between the relative abundances of dominant bacteria genus and soil physicochemical parameters.

Bacterial Genus Taxa	AK	AP	TN	AN	SOM	TK	PH	TP
*Gaiellales*	−0.2509	−0.55635	−0.66434 *	−0.59474 *	0.05236	0.71454 **	0.47552	0.3986
*Vicinamibacterales*	−0.03887	−0.06338	0.27972	0.39295	0.3104	−0.37128	−0.2028	−0.0979
* Gemmatimonadaceae *	0.55834	0.28874	0.58741 *	0.6443 *	−0.03366	−0.35377	−0.61538 *	−0.4965
* KD4-96 *	0.2403	0.1831	0.45455	0.6089 *	0.18325	−0.47986	−0.36364	−0.23776
* Pseudarthrobacter *	0.523	0.33452	0.62238 *	0.49561	−0.32536	−0.43433	−0.56643	−0.31469
* IMCC26256 *	−0.10601	−0.43663	−0.43357	−0.34693	0.00374	0.67601 *	0.27972	0.25175
* Pseudolabrys *	−0.45232	−0.77115 **	−0.79021 **	−0.67616 *	0.17203	0.74956 **	0.65035 *	0.57343
* Vicinamibacteraceae *	0.21909	0.10564	0.45455	0.5735	0.19821	−0.49037	−0.36364	−0.25874
* Gemmatimonas *	−0.13782	−0.63734 *	−0.6014 *	−0.56642	−0.01496	0.55692	0.52448	0.41259
* Ellin6067 *	0.20496	−0.31691	0.15385	0.32215	−0.09349	0.32224	−0.24476	0.38462
* Bradyrhizobium *	0.02474	−0.53875	−0.45455	−0.37525	−0.09723	0.49737	0.46154	0.34266
* SC-I-84 *	0.68202 *	0.60917 *	0.83217 ***	0.84254 ***	−0.19073	−0.40981	−0.88811 ***	−0.48951
* MB-A2-108 *	−0.12722	0.04225	−0.00699	0.13098	0.26179	0.18214	−0.06993	−0.08392
* A4b *	0.18022	0.2817	0.4965	0.45313	−0.00374	−0.61296 *	−0.33566	−0.34266
* Marmoricola *	0.64668 *	0.72185 **	0.86014 ***	0.86024 ***	−0.0748	−0.60596 *	−0.86713 ***	−0.70629 *

Note: * 0.05 (one-tailed); ** 0.01 (one-tailed). *** 0.001 (one-tailed).

**Table 7 microorganisms-12-01190-t007:** Person’s rank correlations between the relative abundances of dominant fungus genus and soil physicochemical parameters.

Fungal Genus Taxa	AK	AP	TN	AN	SOM	PH	TP	TK
* Mortierella *	0.42759	0.32043	0.41259	0.37879	−0.25431	−0.34965	−0.18881	−0.38879
* Gibberella *	−0.37105	0.04225	−0.53846	−0.55225	0.11219	0.39161	0.1049	0.23468
* Tausonia *	0.77743 **	−0.17958	0.23077	0.0885	−0.69934 *	−0.22378	0.01399	−0.05604
* Pseudogymnoascus *	0.21556	0.22184	0.42657	0.32569	−0.00748	−0.52448	−0.32168	−0.36077
* Talaromyces *	−0.04241	0.42607	0.09091	0.04248	0.28422	−0.05594	−0.6014 *	−0.38879
* Pseudeurotium *	0.41698	0.51762	0.6993 *	0.60182 *	−0.19821	−0.59441 *	−0.31469	−0.49037
* Schizothecium *	0.01414	0.26057	0.32867	0.36463	0.3665	−0.43357	−0.44056	−0.46235
* Cystofilobasidiales *	0.33924	0.57044	0.55944	0.46021	−0.07106	−0.60839 *	−0.35664	−0.45184
* Basidiomycota *	−0.36108	−0.59084 *	−0.669 *	−0.42024	0.27536	0.61997 *	0.46935	0.64211 *
* Cephalotrichum *	0.25797	0.51762	0.56643	0.38587	−0.11593	−0.44056	−0.46853	−0.72504 **
* Fusarium *	−0.40992	−0.3979	−0.58042 *	−0.25843	0.3852	0.46853	0.44755	0.45534
* Pseudombrophila *	0.11308	0.3486	0.6014 *	0.59828 *	0.08228	−0.29371	−0.37762	−0.78109 **
* Halenospora *	−0.31206	−0.34276	−0.54035	−0.47782	0.20453	0.48421	0.33334	0.44464
* Peziza *	0.65194 *	0.34105	0.52143	0.57691*	−0.25011	−0.41392	−0.41392	−0.21809
* Sordariomycetes *	−0.64668 *	−0.30282	−0.34965	−0.30445	0.35528	0.25874	0.27972	0.25919

Note: * 0.05 (one-tailed); ** 0.01 (one-tailed).

## Data Availability

The original contributions presented in the study are included in the article, further inquiries can be directed to the corresponding authors.

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
