# Peer review of "The Effects of Exogenous Benzoic Acid on the Physicochemical Properties, Enzyme Activities and Microbial Community Structures of Perilla frutescens Inter-Root Soil"

_microorganisms, 2024, doi:10.3390/microorganisms12061190_

Round 1

Reviewer 1 Report

Comments and Suggestions for Authors

Revision manuscript “The effects of exogenous benzoic acid on the physicochemical properties, enzyme activities and microbial community structures of Perilla frutescens inter-root soil”
Line 4. Please use italic letters to the scientific name Perilla frutescens.
Line 13. Please adjust the abstract as authors guide indicate 200 words, the abstract have more than 250 words and add the acronym BA to benzoic acid since authors guide indicate.
Line 14. Could you please referee to Perilla specie P. frutescens.
Please add the graphical abstract as authors guide indicate.
Line 37. Please use italic letter in Perilla genus.
Line 39. Please use italic letter in Perilla genus or indicate that this is the common name to give the lector the knowledge and use lowercase letter in various.
Line 55. Please add the acronym (BA) as indicate authors guide it should place the first time it appears in the abstract, body of the text and graphs or figures.
Line 58. As a suggestion use BA.
Line 120. As a suggestion use the word donation instead gift.
Line 122. Please indicate how many replicates were used in each evaluated parameter.
Lines123-126. Please briefly indicate something like this concentration were chose according to previous study in our laboratory to justify the use of those concentrations.
Line 126. What do the authors refer to with the term parallels?
Line 127. Where the growing conditions of the plants controlled in the green house? If so please indicate in the section.
Line 132. Could you please add briefly the conventional procedure used in field production.
Lines 146-148. Please add the electronic address of this standard methods and add the correspondent bibliography.
Line 157. Is this the correct cite? please review it. Boyen et al. 2009.
Line 202. Could you please indicate in a brief paragraph the results of quality parameters of plants after 120 days of growth?.
Table 1. As indicate authors guide, please indicate the significance or CK, B30, B60 and B90 at the table bottom.
Line 216. I consider more appropriate to indicate the number of replicates that were carried out in each evaluated parameter and at bottom of the tables just indicate that the value is the average of 3 determinations.
Line 219. Use italic letters in Perilla frutescens.
Line 239. Could you remove microorganisms it is redundant if you are indicating bacteria as a specific microbial group.
Line 248. Maybe you can write similar between treatments with benzoic acid.
Line 259. Please use italic letter in scientific names.
Line 265. Please use italic letter in scientific names.
Line 271. I suggest table 4, as can be seen the Chao 1 index…..
Please consider the alignment in all the tables
Please use the italic letters in scientific names in lines 366, 369, 373, 491, 550, 555, 575,577, 578, 579, 582, 584, 606, 629, 634, 683.
Lines 332-334. Delete italic letter in spp.
Line 336. Please change the order Figure 4 (d, e, f).
Line 393. Please put Pseudolabris instead Pseudolabs.
Line 394 Gemmatimonadaceae is a family, delete the italic letter please.
Line 453. Please add some references to support your statement.
Line 484. Please use subscript in PO4 and superscript in 3-.
Line 508. Please use subscript in H2O formula.
Line 619-621. Please check this statemen Talaromyces is not a type of mycobacterial virus. Trichoderma is not a family, is a genus. The article of Hassan et al 2024 written about the partitivirus founded in Talaromyces pinophyllum these fungi belong to the family Trichocomaceae.
Line 643. Fusarium.
Line 645. Pseudogymnoascus.
Line 699. I believe that H.S acronym have a mistake and it could be H.L, acronym corresponding to the author Li Hui, and the acronym F.L, could be the correspondence to Fang Yuxin author? check please.
Line 704. Persons Yuxin Fang and Chongwei Li are part of the authors of this manuscript? or there is a mistake in the name of the authors?
Line 723. Follow the instructions of authors, use abbreviation for journals.

Reviewer 2 Report

Comments and Suggestions for Authors

The introduction is well done, with an adequate literature review, however the hypothesis should be clearer. The methods section needs to be better described. I do not see a great contribution from the quantification of total forms of nutrients to the scope of the work. Moreover, they do not reflect availability. Additionally, it is not appropriate to apply qualitative statistics to quantitative factors, as performed in the work. As this compromises the results, discussion, and conclusions, I have no comments for the rest of the text.

Comments on the Quality of English Language

In my view, the English is adequate. However, as I’m not a native English speaker, I recommend seeking additional evaluation from the editors.

Author Response

Please refer to the attachment, which can be found here.

Reviewer 3 Report

Comments and Suggestions for Authors

General comments

The authors added three concentrations of benzoic acid to Perilla frutescens cultivated soil and investigated soil physicochemical properties, enzyme activities, and bacterial and fungal community structure. The results were combined to discuss the effects of benzoic acid addition on the biological and non-biological properties of the soil. The authors investigated the effects of benzoic acid addition from several perspectives and found several significant effects.

This manuscript contains information that should be published as a scientific paper. However, it was concluded that the following points need to be revised before it can be published. Especially, the proper interpretation of the results and the organization of the discussion chapters should be rechecked again by all authors.

Major comments

1. The main conclusion of the abstract (L31-32, demonstrating its self-toxic effect on perilla growth and establishing a theoretical foundation for solving the obstacles relating to the continuous cultivation of perilla.) is unclear as to which part of the text it refers to. The authors should either revise the abstract to match the content of the main text or clearly state in the main text the discussion that corresponds to this conclusion. (See also comment 10)

2. The results of the growth of Perilla frutescens need to be presented. Since the results were not shown, it is not possible to determine if part of the description is appropriate.

3. L159-172, The method of the community structure analysis of fungi also needs to be described.

4. L240-243, The logical development of these sentences is not appropriate because the species composition cannot be inferred from the number of OUT itself. The authors need to rewrite the logic of these sentences to be appropriate.

5. L243-246, 16S rDNA analysis does not reveal the community structure of fungi. I assume that the authors probably analyzed the ITS region of 18-26S rDNA. The authors need to understand the experimental method and its principle and describe it correctly.

6. L278-279, The analysis of diversity indexes does not allow to indicate abundance. Therefore, the authors need to revise the sentence appropriately.

7. L290-292, The results of the PCOA of the bacterial community structure of three replicates of B50 (Fig 2a) were not consistent with this sentence. Therefore, the authors need to revise the sentence appropriately.

8. L305-306, As stated by the authors (L302-304), the percentage of Actinobacteriota in CK was lower than the percentage in B30. Therefore, the authors need to revise the text appropriately.

9. L432-669, The content in the discussion chapter is redundant and unclear. The authors should describe the discussion of this study clearly and straightforwardly. The authors need to reorganize the entire discussion chapter to better suit the discussion of this study. Some of the points I noticed are as follows.

L439-441 and L443-445 have the same content.

There are many descriptions of general findings (L475-482, L496-512, L557-566, 597-608, L613-621) and the relevance of these to the results and discussion of this study is unclear. Previous findings and articles should be referred when necessary to the discussion of this study.

L531-543, This study did not examine microbial biomass and therefore cannot be directly compared to these previous findings.

L621, It is unclear which bacteria are meant by "beneficial bacterial genera". This paragraph seems to be discussing fungi.

10. L681-685, I understood that AP, AN and TN affected the composition of the microbial community structure, but I did not understand why this would lead to the conclusion that there is no direct relationship between benzoic acid and microbial community structure. I also felt it contradicted what was stated in L528-539. Instead of stating it in the conclusion at the end, the authors should state the explanation that leads to this conclusion in the discussion chapter.

Minner comments

L39, Word(s) seem to be missing at the end of this sentence ("Perilla seeds are rich in α -.").

All tables and figures, Titles are needed. The current text is a description of the figures and tables.

L241 and L591, "bacterial and microbial" should be "bacterial and fungal".

L234 and L244, 16S DNA should be 16S rRNA gene.

L297 and L310, "gate" should be corrected to "phylum".

Figure 4, 5, 6, 7, 8, and 9, The letters in the figures need to be enlarged.

L385, Figure 6 seems to be a mistake for Figure 7a. It would be better to add "a" or "b" as necessary throughout the manuscript.

L440 and L445, If "This" refers to the previous sentence, the logical development is not appropriate. The sentence needs to be revised so that the preceding and following sentences are correctly connected.

L615, Since the sentence seems to describe the results of this study, I think "increased" is correct, not "increase".

LL644, "ium" seems to be a typo.

I hope these comments will be helpful.

Author Response

(The authors gave the same response as above.)

Round 2

Reviewer 1 Report

Comments and Suggestions for Authors

Dear author,

Thank you for your time and consideration of the comments, which have been fully taken into account.

Yours sincerely

Author Response

Dear Reviewer,

Thank you for your helpful suggestion. I think it is very important for me. Thank you very much for taking the time to review our article. We have received the second round of review report and made the necessary modifications.

Best wishes,

Yours Sincerely

Tongtong Xue

Reviewer 2 Report

Comments and Suggestions for Authors

An unresolved issue in statistics, as presented in our initial review, can indeed impact the results, discussion, and conclusion.

Comments on the Quality of English Language

.

Reviewer 3 Report

Comments and Suggestions for Authors

The authors have revised the manuscript in response to most of the reviewers' comments. The following notational errors should be corrected.

L244. “diagram,and” (A space is required between two words)

L249, “diagrams.A total”

L296, “bac-terial” (Unnecessary hyphen)

L297, “com-munity”

I hope my comments will help the authors to complete the manuscript.
